# Secretoneurin as a Novel Biomarker of Cardiovascular Episodes: Are We There Yet? A Narrative Review

**DOI:** 10.3390/jcm11237191

**Published:** 2022-12-03

**Authors:** Jiří Plášek, Marie Lazárová, Jozef Dodulík, Patrik Šulc, David Stejskal, Zdeněk Švagera, František Všianský, Jan Václavík

**Affiliations:** 1Dept. of Internal Medicine and Cardiology, University Hospital Ostrava, 70800 Ostrava, Czech Republic; 2Research Center for Internal and Cardiovascular Diseases, Faculty of Medicine, University of Ostrava, 70300 Ostrava, Czech Republic; 3Institute of Laboratory Medicine, University Hospital Ostrava, 70800 Ostrava, Czech Republic; 4Institute of Laboratory Medicine, University of Ostrava, 70103 Ostrava, Czech Republic

**Keywords:** secretoneurin, ryanodine receptor, heart failure, arrhythmogenesis, cardiac arrest, CaMKII

## Abstract

Secretoneurin (SN) is a 33 amino-acid evolutionary conserved neuropeptide from the chromogranin peptide family. SN’s main effects may be cardioprotective and are believed to be mediated through its inhibition of calmodulin-dependent kinase II (CaMKII), which influences intracellular calcium handling. SN inhibition of CaMKII suppresses calcium leakage from the sarcoplasmic reticulum through the ryanodine receptor. This action may reduce the risk of ventricular arrhythmias and calcium-dependent remodelling in heart failure. SN is also involved in reducing the intracellular reactive oxygen species concentration, modulating the immune response, and regulating the cell cycle, including apoptosis. SN can predict mortality in different disease states, beyond the classical risk factors and markers of myocardial injury. Plasma SN levels are elevated soon after an arrhythmogenic episode. In summary, SN is a novel biomarker with potential in cardiovascular medicine, and probably beyond.

## 1. Introduction

Secretoneurin (SN) is a novel biomarker that has shown potential in predicting adverse arrhythmic events and the progression of heart failure. SN acts through a cellular pathway that is different from the natriuretic peptide and troponin pathways. Its main effects are believed to be mediated by calmodulin-dependent-kinase II (CaMKII); however, other up- or down-stream cellular pathways may be involved [1,2].

CaMKII is a regulator of calcium handling inside the cell. Calcium is central to excitation-contraction coupling and intracellular signalling. Loss of proper calcium cycling is a known surrogate of heart disease [3]. Thus, inhibiting CaMKII may have protective effects on the diseased myocardium [3]. Myocardial CaMKII inhibition improved contraction and suppressed arrhythmias by suppressing calcium (Ca^2+^) leaks from the sarcoplasmic reticulum (SR) in both animal models and the failing human myocardium [3,4]. SN is probably up-regulated as a compensatory mechanism linked to abnormal calcium regulation [3].

Therefore, SN is worth studying in different disease states. Here, we review the recent literature regarding SN. In considering the evidence, we should keep in mind what an ideal marker should look like. In our opinion, a useful biomarker should: Provide independent information about the risk and prognosis of the studied disease;Account for a significant proportion of the identified risk;At best, be useful for stratifying the disease into clinically relevant categories;Be easily reproducible with low inter-sample variation, high sensitivity, and high specificity.

## 2. Basic Properties of Secretoneurin

SN ranges from 31 (zebrafish) to 43 (lamprey) amino acids in length. SNs are evolutionarily conserved neuropeptides, mostly 33 (human, chicken, and frog) amino acids long, which arose in tetrapods, and they vary in very few positions [5]. All known SN peptides contain the YTPQ-X-LA-X(7)-EL sequence, which constitutes its central core [5]. In addition, SNa and SNb paralogs have been identified in teleosts, and the N-terminus and the middle of teleost SNa are identical to the mammalian sequence [6]. In contrast, teleost SNb, as compared to mammalian SNb, are only moderately conserved [6].

SN belongs to the secretogranin/chromogranin peptide family [5,6]. SN is produced by the proteolytic cleavage of a precursor protein, secretogranin II (SgII, also known as chromogranin C, 617 amino acids), which occurs in several tissues, including cardiomyocytes and neuroendocrine tissues [7]. SgII processing is known to be species- and tissue-specific; thus, sometimes larger SN immunoreactive fragments are generated, in addition to free SN. Most studies have supported the notion that the proprotein convertase subtilisin kexin-1 plays a central role in the generation of SN from SgII [8]. Exact sequence analysis of SgII can be found at https://www.uniprot.org/uniprotkb/P13521/entry#sequences (accessed on 1 December 2022).

Granin proteins are essential for protein sorting and hormone packaging in the regulated secretory pathway, where proteins cross the Golgi apparatus to the trans-Golgi network. Under acidic, high-calcium conditions, SgII may be processed by prohormone convertases in the secretory vesicle. Secretory vesicles that carry SN fuse with the cell membrane, and SN is released through calcium-dependent exocytosis. Once released from the cell, it binds to the SN receptor in an autocrine, paracrine, or neuroendocrine manner [9].

## 3. Biological Effects of Secretoneurin

In general, granins are predominantly known for their calcium-binding abilities. Granins have been extensively studied as neuroprotective factors and as co-factors of local immune and inflammatory reactions [10]. Before SN was identified as a potential biomarker of heart failure, acute coronary syndrome, and arrhythmogenesis, the prognostic values of chromogranin A and B were studied in similar conditions with variable results. In particular, chromogranin A is an independent predictor of long-term mortality and heart failure-related hospitalization in patients with acute coronary syndromes [11]. However, in patients with heart failure, chromogranin A did not show any additional value over contemporary markers [12]. In contrast, chromogranin B is up-regulated in chronic heart failure, and its plasma levels are correlated with disease severity [13].

SN has been detected in non-negligible amounts (range: 20–1500 pmol/L) in human serum, cerebrospinal fluid, and urine [14]. SN has also been detected in human dental pulp and afferent nerve fibres that are frequently associated with blood vessels [10,15]. SN immunoreactivity was also detected in the unmyelinated C-fibers which transmit predominantly nociceptive impulses [15]. All the biological effects of SN are not completely understood. We can only speculate that the effects of SN may be exerted by SN receptors on the cell membrane. There is evidence that the SN receptor is a G-coupled membrane receptor [16]. In addition, SN may also act directly at different subcellular levels. The biological effects of SN are summarized in Figure 1. 

In concentrations 10x the plasma concentration, SN inhibits CaMKII by 20–30%. SN exerts its effect by binding to calmodulin [3]. A direct effect, without binding to calmodulin, is thus again highly speculative. SN reduces ryanodine receptor (RyR)-mediated Ca^2+^ leakage by inhibiting RyR phosphorylation. This activity leads to a reduced risk of ventricular arrhythmia [4]. Furthermore, sarcoplasmic Ca^2+^ leakage leads to calmodulin calcification, which binds and activates CaMKII; thus, SN may also indirectly inhibit CaMKII at plasma concentrations that correspond to the levels observed in heart disease [4].

Interestingly, SN may also exert an anti-apoptotic effect on endothelial cells. This SN effect could be initiated by the activation of phosphoinositide-3 kinase/protein kinase B and the mitogen-associated protein kinase (MAPK) pathway [17]. In human umbilical vein endothelial cells, SN promoted proliferation and chemotaxis and reduced apoptosis [18]. In a myocardial infarction model, SN also acted as an endogenous stimulator of vascular endothelial growth factor and induced angiogenesis [19].

Of note, in animal models SN also enhances bone regeneration by increasing blood vessel and bone marrow formation [20].

In contrast to its beneficial roles, SN may also trigger the selective migration of monocytes and fibroblasts. This activity could potentially enhance inflammatory responses that lead to the acceleration of atherosclerosis [21].

## 4. Cellular Pathophysiology of Secretoneurin

Oxidative stress is a significant component of the cellular and subcellular processes that lead to inflammation and compensatory mechanisms, such as cell hypertrophy, fibrosis, and/or apoptosis. Reactive oxygen species (ROS) regulate mitogenic signal transduction in cardiomyocytes (evidenced in an animal experiment). Consequently, ROS may modulate hypertrophic signalling through MAPK pathways [22]. At the cellular and molecular levels, hypertrophic adaptions are quite complex in nature; they include paracrine, autocrine, and circulating biologically active effectors that react to many stimuli. Among these stimuli, the most important is wall stress [23].

Myocardial stretch triggers a chain of events: (1) the release of angiotensin II; (2) the release/formation of endothelin; (3) the activation of NADPH oxidase and transactivation of endothelium growth factor; (4) the production of mitochondrial ROS; (5) the activation of redox-sensitive kinases; (6) an increase in the intracellular sodium concentration; and (7) a transient increase in the Ca^2+^ amplitude through the Na+/Ca^2+^ exchanger. SN was found to attenuate isoproterenol-induced myocardial cell hypertrophy by suppressing ROS production and inducing the activities of superoxide dismutase and catalase [24]. These effects are known to be mediated by adenosine-monophosphate or the extracellular-regulated kinase (ERK)/MAPK pathway. It is less probable that SN directly affects myocardial cell hypertrophy [20]. Nevertheless, SN gene therapy (in an animal model of isoproterenol-induced hypertrophy) led to significant reductions in natriuretic peptide levels, the heart-weight to body-weight ratio, and surprisingly, interstitial fibrosis [25].

Other results have indicated that SN, both acutely and after prolonged exposure, induced the augmented expression of endothelial nitric oxide synthase and calmodulin [25]. In this respect, SN may play a protective role in ischaemic heart disease by enhancing endothelium-dependent relaxation. Endothelium-dependent relaxation may also involve the activation of cyclooxygenase, which would lead to the production of prostacyclin [25].

SN may be implicated in the inflammatory response, because it modulates leukocytes, endothelial cells, and mesenchymal cells. SN promotes transendothelial leukocyte migration and adhesion; this activity is comparable to that promoted by tumour necrosis factor alpha [26]. In response to injury, SN stimulates an increase in the migration and locomotion of human monocytes [27]. In endothelium, SN activates the G-protein-coupled and protein kinase C-mediated signalling pathways. As a result, the cytoplasmatic calcium concentration rises [26]. SN also inhibits the tight junctions between endothelial cells, which increases their permeability [28]. Plasma SN concentrations that led to endothelium-derived relaxation in vitro were higher than those detected in humans with hypoxic conditions [29,30]. However, plasma SN concentrations may not precisely reflect the tissue (endothelium) concentration. The most common pathological conditions that lead to increases in SN expression are the growth of active, predominantly neuroendocrine tumours and hypoxia [13,31].

## 5. Plasma Secretoneurin Determination/Normal Range

Plasma SN measurements have been performed using different methods; e.g., a homemade radioimmunoassay [32], biosensors [33], and a commercially available ELISA (CardiNor AS, Oslo, Norway) [34]. Currently, the predominate method is the ELISA, which was used to determine the reference limits in middle-aged (mean age: 36 years), healthy individuals. The median plasma SN concentrations were 38 pmol/L in women and 33 pmol/L in men [32]. Within- and between-subject variations were 9.8% (95%, CI: 8.7–11%) and 20% (95%, CI: 15.4–28%), respectively; the reference change values were 38.7% (95%, CI: 35.5–42.7%) and −27.9 (95%, CI: −29.9 to −26.2), respectively; and the individuality index was 0.6 (95%, CI: 0.42–0.78) [30]. Plasma SN levels were not influenced by physical activity, particularly bicycle ergometry [4].

## 6. Clinical Utility of Secretoneurin

SN is involved in many processes, including apoptosis, the immune response, inflammation/chemotaxis, endothelium relaxation, calcium handling, arrhythmogenesis, and cell cycle regulation. Given this wide range of biological effects and reactivities, SN has a potentially broad range of clinical utility in cardiology as a biomarker of heart failure/arrhythmogenic risk; as a marker of neuroendocrine cell tumour activity; and as a marker of inflammation. SN was moderately elevated in patients with catecholaminergic polymorphic ventricular tachycardia (CPVT) who had not experienced arrhythmogenic episodes [4]. The same patients were sampled at the same time for N-terminal-pro hormone B-type natriuretic peptide (NT-pro-BNP) and troponin, and both of these markers were in the normal range [4]. An up-regulation of plasma SN levels may be a compensatory mechanism to override CaMKII activation, which leads to RYR2 hyperphosphorylation (hyperactivation), sarcoplasmic Ca^2+^ leakage, and delayed afterdepolarizations (DAD) [3,4] (Figure 2). As a result of Ca^2+^ leakage, sodium–calcium ion exchanger activity is increased. Therefore, early afterdepolarizations may occur [35]. DADs comprise the key mechanism for activating ventricular arrhythmias in patients with CPVT [35]. 

Another clinical situation that has been studied is out-of-hospital cardiac arrest (OHCA) due to arrhythmia. Using sequential plasma sampling, the highest SN plasma levels were observed less than 6 h after an OHCA. In the following 24 h, plasma SN levels dropped to the normal range [4]. The increase in plasma SN values after an arrhythmic episode may be explained by a compensatory hypersecretion/cleavage in response to abnormal calcium metabolism. Alternatively, SN may be overexpressed/oversecreted as a part of the systemic inflammatory response syndrome and the acute-phase protein spillover which occurs after a cardiac arrest. 

In patients with coronary artery bypass grafts (CABG), plasma SN levels were reduced after the procedure in survivors, but not in non-survivors, where the SN levels remained elevated [34]. SN levels were also significantly higher, on average (173 vs. 143 pmol/L), in non-survivors compared to survivors after CABG [36]. Similarly, SN concentrations in critically ill patients were higher in non-survivors than in survivors, irrespective of whether the associated acute respiratory failure (ARF) was related to a cardiovascular condition [35]. The area under the receiver operating curve (AUC) for SN in patients with cardiovascular-related ARF was 0.72 (95%, CI: 0.65–0.79), and the AUC of NT-pro-BNP in the same population was 0.64 (95%, CI: 0.65–0.79) [37].

Preoperative SN plasma levels may provide a better marker for stratifying patients with aortic stenosis to identify those with a higher mortality risk. In a small observation study, plasma SN levels were higher in non-survivors (mean: 156, range: 133–209 pmol/L) than in survivors (mean: 140, range: 116–155 pmol/L) [38]. As suggested previously, in patients undergoing cardiac surgery [32], the SN cut-off value for predicting higher mortality risk was >204 pmol/L, which held also true for a cohort of patients with aortic stenosis [39]. We may only speculate whether SN plasma levels will also be helpful in discriminating different types of heart failure. We believe it will most probably identify those patients with higher arrhythmia and/or mortality risk irrespective of the type of heart failure.

In critically ill patients that were admitted to the intensive care unit with sepsis, plasma SN levels could predict mortality in addition to the classical risk factors, including age [38]. Of note, SN has also been used clinically as a marker for the presence of tumours. In patients with cancer, the plasma SN concentration ranged from 58.3 ± 8.1 to 410 ± 113.9, which was much higher than the average observed in healthy subjects (22.1 ± 1.1 pmol/L) [26,29,34].

## 7. Conclusions

SN seems to be a novel prognostic marker, which adds information beyond that provided using the classical markers (cardiac troponins and brain natriuretic peptides) of cardiac injury and overload. SN may predict the risk of hospital mortality in critically ill patients, in patients after a CABG, and in patients with ARF. In patients with OHCA of an arrhythmogenic cause, SN was elevated for a few hours after the arrhythmogenic episode, and then dropped. In patients with CPVT, SN was moderately elevated at baseline, irrespective of an arrhythmogenic episode. Thus, SN is a novel biomarker that has shown potential for use in cardiovascular medicine, and will probably be a useful biomarker in conditions beyond those involving the cardiovascular system.

## Figures and Tables

**Figure 1 jcm-11-07191-f001:**
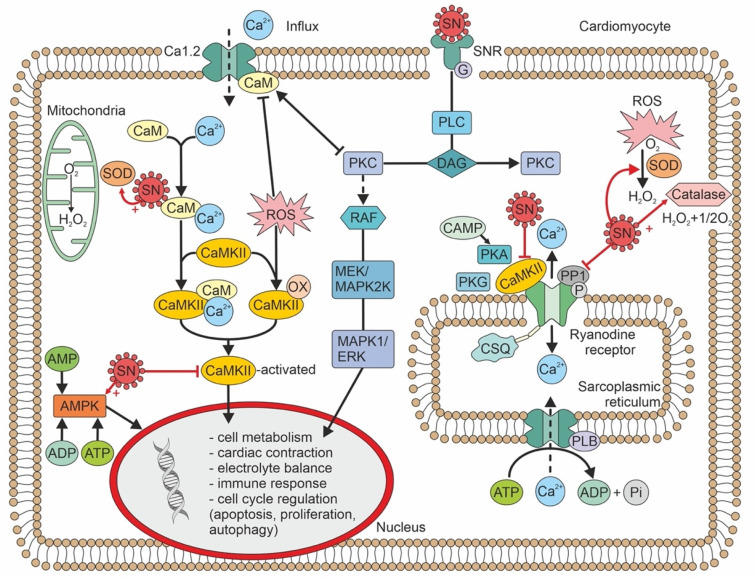
Intracellular signaling pathways of secretoneurin, and biological effects. ADP—adenosine diphosphate; AMP—adenosine monophosphate; AMPK—adenosine monophosphate kinase; ATP—adenosine triphosphate; cAMP—cyclic adenosine monophosphate; CaM—calmodulin; CaMKII—calmodulin-dependent kinase II; CSQ—calsequestrin; DAG—diacylglycerol; ERK—extracellular-regulated kinase; MAPK—mitogen-associated protein kinase; MEK—MAPK/ERK kinase; OX—oxidized; SOD—superoxide dismutase; PLB—phospholamban; P—phosphorus; PKA—protein kinase A; PKC—protein kinase C; PKG—protein kinase G; PLC -phospholipase C; RAF—serine-threonine protein kinase; ROS—reactive oxygen species; SN—secretoneurin; SNR—secretoneurin receptor.

**Figure 2 jcm-11-07191-f002:**
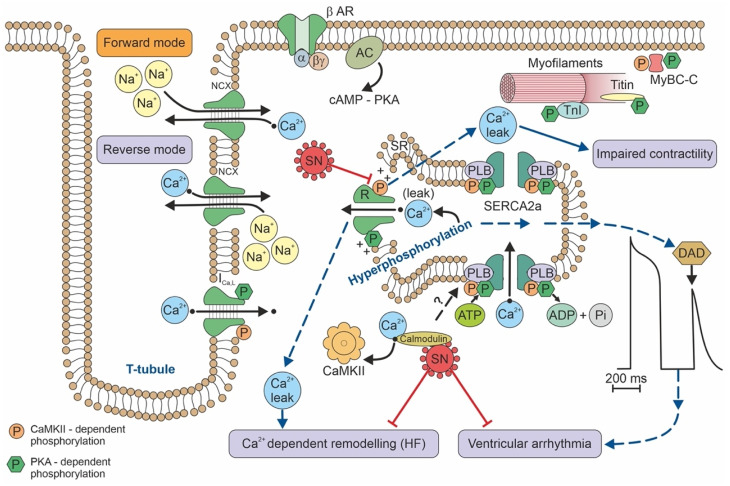
Calcium handling/signaling in a ventricular cardiomyocyte illustrates the potential involvement of SN in arrhythmogenic risk, heart failure, and calcium-dependent remodeling. AC—adenylyl cyclase; ADP—adenosine diphosphate; ATP - adenosine triphosphate; β-AR—beta adrenergic receptor; cAMP—cyclic adenosine monophosphate; CaMKII—calmodulin-dependent kinase II; DAD—delayed afterdepolarizations in the electrocardiogram; HF—heart failure; I_Ca,L_—calcium L-type channel current; MyBP-C—myosin binding protein C1; NCX—sodium-calcium exchanger; P—phosphorus; PKA—protein kinase A; PLB—phospholamban; R—ryanodine receptor; SERCA2a—the sarcoplasmic/endoplasmic reticulum Ca^2+^ ATPase 2a; SN—secretoneurin; SR—sarcoplasmic reticulum; TnI—troponin I.

## Data Availability

Not applicable.

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
