# Peer review of "Secretoneurin as a Novel Biomarker of Cardiovascular Episodes: Are We There Yet? A Narrative Review"

_jcm, 2022, doi:10.3390/jcm11237191_

Round 1
Reviewer 1 Report
Cardiovascular diseases are an important problem of morbidity and mortality in the world, and a burden for the national healthcare systems. Identifying novel markers of prognosis could modify the approach of healthcare professionals.
The manuscript entitled: «Secretoneurin as a novel biomarker of cardiovascular episodes: are we there yet? A narrative review» describes the basis properties and biological effects of secretoneurin, as well as current aspects of the biomarker among patients with different types of cardiovascular diseases. Therefore, the finding of the Clinical utility of secretoneurin as a predictor of the worsened prognosis is crucial.
Conclusion:
It would be interesting to know the authors's viewpoint concerning the perspectives of the secretoneurin in different types of heart failure.
References:
The updating of the references is needed: 67% of the references were published more than 10 years ago
Author Response
We thank the reviewer for his/her valuable suggestions.
We revised the manuscript accordingly to his/her recommendations ,namely:
- We have corrected all the references, since we have to add some new in the very beginning, altogether 4 new references were added
- We have corrected also both the figures, since there were spelling errors as the reviewer pointed out.
- “Basic properties” paragraph was rewritten almost completely
We believe that the article is now significantly improved. The changes are highlighted by the red font in the manuscript. Please find below the corresponding point-by-point response.
#Reviewer 1:
Cardiovascular diseases are an important problem of morbidity and mortality in the world, and a burden for the national healthcare systems. Identifying novel markers of prognosis could modify the approach of healthcare professionals.
The manuscript entitled: «Secretoneurin as a novel biomarker of cardiovascular episodes: are we there yet? A narrative review» describes the basis properties and biological effects of secretoneurin, as well as current aspects of the biomarker among patients with different types of cardiovascular diseases. Therefore, the finding of the Clinical utility of secretoneurin as a predictor of the worsened prognosis is crucial.
Conclusion:
It would be interesting to know the authors's viewpoint concerning the perspectives of the secretoneurin in different types of heart failure.
References:
The updating of the references is needed: 67% of the references were published more than 10 years ago
Author response:
- We added our perspectives on the secretoneurin in different types of heart failure. We have no specific data for patients with reduced and/or preserved ejection fraction to risk stratify any cohort. We may only speculate, that most probably SN may identify patients with higher risk of arrhythmic events and/or mortality, irrespective of the type of the heart failure.
- Reference list was up-dated,4 new references were added as a result; also 2 duplicite references were removed

Reviewer 2 Report
As presented, it is confusing at times and difficult to understand. Even though SN is described as 33-amino acid residue peptide, no where in this review its or its precursor protein sequences are provided. The sequence analyses may provide the reasons why it may be a biomarker. Statement "phylogenetically conserved" needs to be expanded with analysis.
Subheading 2 Basis properties of secretoneurin is confusing. Do they mean Basic properties? Here SN is defined as 31 to 42 amino acids in length, different from what SN is defined as in the beginning of the abstract.
Symbol SgII is introduced without defining earlier. Is Granin SgII? Confusing. on page 7 AUC is described without any representative figure. This caused confusion for me. conclusion section starts with "SN seems to be a relevant marker. Whereas the beginning of Introduction SN is stated as a novel biomarker. Which is correct?
Author Response
We thank the reviewer for his/her valuable suggestions.
We revised the manuscript accordingly to his/her recommendations, namely.
- We have corrected all the references, since we have to add some new in the very beginning, altogether 4 new references were added
- We have corrected also both the figures, since there were spelling errors as the reviewer pointed out.
- “Basic properties” paragraph was rewritten almost completely
We believe that the article is now significantly improved. The changes are highlighted by the red font in the manuscript. Please find below the corresponding point-by-point response.
#Reviewer 2 statements:
- As presented, it is confusing at times and difficult to understand. Even though SN is described as 33-amino acid residue peptide, no where in this review its or its precursor protein sequences are provided. The sequence analyses may provide the reasons why it may be a biomarker.
- Statement "phylogenetically conserved" needs to be expanded with analysis.
- Subheading 2 Basis properties of secretoneurin is confusing. Do they mean Basic properties?
- Here SN is defined as 31 to 42 amino acids in length, different from what SN is defined as in the beginning of the abstract.
- Symbol SgII is introduced without defining earlier. Is Granin SgII? Confusing.
- on page 7 AUC is described without any representative figure. This caused confusion for me.
- conclusion section starts with "SN seems to be a relevant marker. Whereas the beginning of Introduction SN is stated as a novel biomarker. Which is correct?
Author response:
- We added the number of amino acids of SN precursor, the Secretogranin II (SgII) and the link, where secretogranin II sequence analysis can be found.
- We expanded the analysis of evolutionary conservation of the SN amino acid structure adding most conserved core amino-acid sequence.
- We corrected to Basic as you rightly proposed
- SN ranges from 31 (zebrafish) to 41 (shark), 33 in human. We added this information under this subheading. In the abstract we use the human SN length.
- Secretogranin II is introduced now at page 2, line 60
- The area under the receier operating curve (ROC) is expressed as a numerical comparison with respective confidence intervals, figure cannot be reproduced from the original paper due to the copyright. We believe that numerical comparison is informative enough, we added also the explanation that the ROC is meant.
- We stick to „novel“ since the title also contains this wording, the conclusion is corrected as you recommended.

Round 2
Reviewer 1 Report
The manuscript was really updated due to all comments.